# Object-Centric Representation Learning with Generative Spatial-Temporal Factorization

**Li Nanbo**
School of Informatics
University of Edinburgh
nanbo.li@ed.ac.uk

**Muhammad Ahmed Raza**
School of Informatics
University of Edinburgh
m.a.raza@ed.ac.uk

**Hu Wenbin**
School of Informatics
University of Edinburgh
wenbin.hu@ed.ac.uk

**Zhaole Sun**
School of Informatics
University of Edinburgh
zhaole.sun@ed.ac.uk

**Robert B. Fisher**
School of Informatics
University of Edinburgh
rbf@inf.ed.ac.uk

## Abstract

Learning object-centric scene representations is essential for attaining structural understanding and abstraction of complex scenes. Yet, as current approaches for unsupervised object-centric representation learning are built upon either a stationary observer assumption or a static scene assumption, they often: i) suffer single-view spatial ambiguities, or ii) infer incorrectly or inaccurately object representations from dynamic scenes. To address this, we propose *Dynamics-aware Multi-Object Network* (DyMON), a method that broadens the scope of multi-view object-centric representation learning to dynamic scenes. We train Dy-MON on *multi-view-dynamic-scene* data and show that DyMON learns—without supervision—to factorize the entangled effects of observer motions and scene object dynamics from a sequence of observations, and constructs scene object spatial representations suitable for rendering at arbitrary times (*querying across time*) and from arbitrary viewpoints (*querying across space*). We also show that the factorized scene representations (w.r.t. objects) support querying about a single object by space and time independently.

## 1 Introduction

Object-centric representation learning promises improved interpretability, generalization, and data-efficient learning on various downstream tasks like reasoning (e.g. [18, 39]) and planning (e.g. [30, 2, 41]). It aims at discovering compositional structures around objects from the raw sensory input data, i.e. a *binding problem* [12], where the *segregation* (i.e. factorization) is the major challenge (e.g. [9, 4]), especially in cases of no supervision. In the context of visual data, most existing focus has been on single-view settings, i.e. decomposing and representing 3D scenes based on a single 2D image [1, 10, 26] or a fixed-view video [23]. These methods often suffer from single-view spatial ambiguities and thus show several failures or inaccuracies in representing 3D scene properties. It was demonstrated by Nanbo et al. [31] that such ambiguities could be effectively resolved by multi-view information aggregation. However, current multi-view models are built upon a foundational static-scene assumption. As a result, they: 1) require static-scene data for training and 2) cannot handle well dynamic scenes where the spatial structures evolve over time. This greatly harms a model's potentials in real-world applications.

In this work, we target an unexplored problem—unsupervised object-centric latent representation learning in *multi-view-dynamic-scene* scenarios. Despite the importance of the problem to spatial-

temporal understanding of 3D scenes, solving it presents several technical challenges. Consider one particularly interesting scenario where both an observer (e.g. a camera) and the objects in the scene are moving at the same time. To aggregate 3D object information from two consecutive observations, an agent needs not only to handle the cross-view object correspondence problem [31] but also to reason about the independent effects of the scene dynamics and observer motions. One can consider the aggregation as a process of answering two questions: "how much has an object really changed in the 3D space" and "what previous spatial unclarity can be clarified by the current view". In this paper, we refer to the relationship between the scene spatial structures and the viewpoints as the *temporal entanglement* because the temporal dependence of them complicates the identification of the *independent generative mechanism* [35].

We introduce DyMON (***D**ynamics-aware **M**ulti-**O**bject Network*), a unified unsupervised framework for multi-view object-centric representation learning. Instead of making a strong assumption of static scenes as that in previous multi-view methods, we only make two weak assumptions about the training scenes: i) observation sequences are taken at a high frame rate, and ii) there exists a significant difference between the speed of the observer and the objects (see Sec. 3). Under these two assumptions, in a short period, we can transition a *multi-view-dynamic-scene* problem to a *multi-view-static-scene* problem if an observer moves faster than a scene evolves, or to a *single-view-dynamic-scene* problem if a scene evolves faster than an observer moves. These local approximations allow DyMON to learn independently the generative relationships between scenes and observations, and viewpoints and observations during training, which further enable DyMON to address the problem of scene spatial-temporal factorization, i.e. solving the observer-scene *temporal entanglement* and scene object decomposition, at test time.

Through the experiments we demonstrate that: **(i)** DyMON represents the first unsupervised multi-view object-centric representation learning work in the context of dynamic-scene settings that can train and perform object-oriented inference on *multi-view-dynamic-scene* data (see Sec. 5). **(ii)** DyMON recovers the *independent generative mechanism* of an observer and scene objects from observations and permits querying predictions of scene appearances and segmentations across both space and time (see Sec. 5.1). **(iii)** As DyMON learns scene representations that are factorized in terms of objects, DyMON allows single-object manipulation along both the space (i.e. viewpoint) and time axis—e.g. replays dynamics of a single object without interferring the others (see Sec. 5.1).

## 2   Background

**Object-centric Representations** Consider object-centric representation inference as the inverse problem of an observation generation problem (i.e. the *vision-as-inverse-graphics* [40] idea). In the forward process, i.e. observation generation, we have a scene well-defined by a set of parameter vectors $\mathbf{z} = \{z_k\} = \{z_1, z_2, ..., z_K\}$, where a $z_k \in \mathbb{R}^D$ specifies one and only one object in the scene. An observation of the scene $\mathbf{x}$, e.g. an image $x \in \mathbb{R}^M$ or an RGB image $x \in \mathbb{R}^{M \times 3}$, can be taken only by a specified observer (often defined as $v \in \mathbb{R}^d$) which is independent of the scene in the forward problem, using a specific mapping $g : \mathbb{R}^D \times \mathbb{R}^d \mapsto \mathbb{R}^{M \times 3}$. Assuming a deterministic process, an observation $x$ is generated as $x = g(\mathbf{z}, v)$, where $v$ is often omitted in single-view scenarios (e.g. [1, 10]). With the forward problem defined, we can describe the goal of learning an object-centric representation as inferring the intrinsic parameters of the objects $\{z_k\}$ that compose a scene $\mathbf{z}$ based on the scene observation $\mathbf{x}$. In other words, computing a factorized posterior $p(\mathbf{z}|\mathbf{x}) = p(z_1, z_2, ..., z_K|\mathbf{x})$, even though it is computationally intractable. As the number of objects is unknown in the inverse problem, it is worth noting that i) $K$ is often set globally to be a sufficiently large number (greater than the actual number of objects) to capture all scene objects, and ii) we allow empty "slots".

**Temporal Entanglement** The dynamic nature of the world suggests that the spatial configuration of a scene (denoted by $\mathbf{z}^t$) and an observer $v^t$ are bound to the specific time $t$ that an observation is taken (i.e. $x^t = g(\mathbf{z}^t, v^t)$). Let $\mathbf{X} = \{(x^t, v^t)\}_{1:T}$ [1] represent a data sample, e.g. a sequence or set of multi-view image observations, from dataset $\mathcal{D}$, where $T$ is the number of the images in the sample. Assuming $\mathbf{z}^t$ is given in the data sample for now, i.e. focusing on the generative process only, we augment a scene data sample as $\mathbf{X}_a = \{(x^t, v^t, \mathbf{z}^t)\}_{1:T}$. In general, we assume an independent scene-observer relation: $\mathbf{z}^t \perp v^t | \emptyset$ but they nevertheless become dependent when the

---

[1]We define $(\cdot)$ as a joint sample indicator that forbids independent sampling of the random variables wherein.

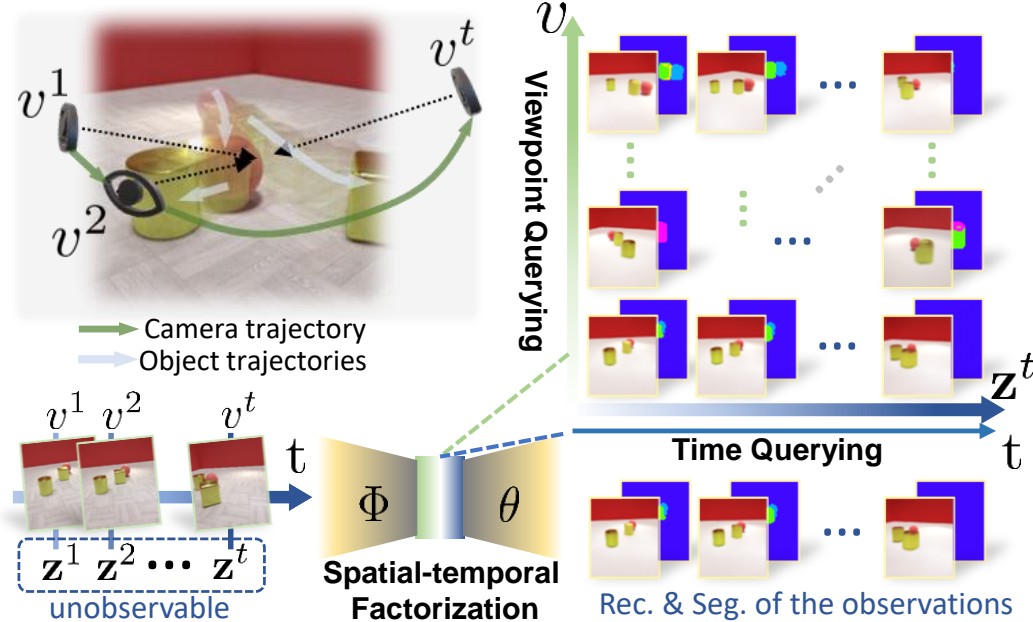

Figure 1: **Top Left:** *Multi-view-dynamic-scene* setup. $v$ with a time index superscript denotes the spatial configuration (e.g. position, orientation, etc.) of an observer at a specific time. We highlight one particular interesting, yet unexplored, scenario where both an observer and scene objects are moving at the same time—which entangles the independent effects of the observer's and scene objects' motions on an scene observation, an image sequence (see **bottom left**). A latent variable $\mathbf{z}$ that is indexed by time describes the objects and their spatial configuration at a specific time (See Sec. 2 for detailed definition). **Right:** DyMON decouples the generative effects of observer motions and scene object motions and enables: 1) reconstruction and factorization of the observed views (see **bottom right**), and 2) novel-view appearance and decomposition prediction for arbitrary times—querying across both space and time (see **top right**).

corresponding observation is given: $\mathbf{z}^t \not\perp v^t | x^t$. Under a static-scene assumption, we can treat an augmented data sample as $\mathbf{X}_a = \{(x^t, v^t), \mathbf{z}^t\}_{1:T}$ where $\mathbf{z}^t$ and $v^t$ are separable (i.e. can be sampled independently). In this case, to recover the independent generative mechanism (i.e. train a $g(\cdot)$) w.r.t. scenes and observers from data, GQN [8] and MulMON [31] fix $\mathbf{z}^t$ to $\mathbf{z}$ and intervene on the viewpoints $v^t$. From a causal perspective, this can be seen as estimating $p(x^{t'} | \boldsymbol{do}(v^t = v^{t'}), \mathbf{z}^t = \mathbf{z})$, where $(x^{t'}, v^{t'}) \sim \{(x^t, v^t)\}_{1:T}$, implicitly under a causal model: $\mathbf{z}^t \rightarrow x^t \leftarrow v^t$. However, in dynamic settings, the same estimation, i.e. sampling $(x^{t'}, v^{t'}) \sim \{(x^t, v^t)\}_{1:T}$ independently of $\mathbf{z}^t$, is forbidden by the $(\cdot)$ indicator. Intuitively, an observer cannot take more than one observations from different viewpoints at the same time $t$. In this paper, we refer to this issue as *temporal entanglement* in view of the temporal implication of the $(\cdot)$ indicator.

## 3 DyMON

Our goal is to train a multi-view object-centric representation learning model that recovers the *independent generative mechanism* of scene objects and their motions and observer motions from dynamic-scene observations. In this section, we detail how DyMON addresses these two presented challenges: 1) temporal disentanglement (see Sec. 3.1), and 2) scene spatial factorization (see Sec. 3.2). We discuss the training of DyMON in Sec. 3.3.

### 3.1 Temporal Disentanglement

The key to resolving *temporal entanglement*, i.e. temporal disentanglement, is to enable sampling $(x^t, v^t)$ independently of $\mathbf{z}^t$, or $(x^t, \mathbf{z}^t)$ independently of $v^t$. This is seemingly impossible in the *multi-view-dynamic-scene* setting as it requires to fix either $\mathbf{z}^t$ (static scene) or $v^t$ (single-view) respectively. In this paper, we make two assumptions about the training scenes to ensure the satisfaction of the

aforementioned two requirements without violating the global *multi-view-dynamic-scene* setting. Let us first describe the dynamics of scenes and observers with two independent dynamical systems:

$$\mathbf{z}^{t+\Delta t} - \mathbf{z}^t = \overline{f_{\mathbf{z}}}(\mathbf{z}^t, t)\Delta t \quad , \quad v^{t+\Delta t} - v^t = \overline{f_v}(v^t, t)\Delta t, \tag{1}$$

where $t$ and $t + \Delta t$ are the times that two consecutive observations were taken, $\overline{f_{\mathbf{z}}}(\mathbf{z}^t, t)$ and $\overline{f_v}(v^t, t)$, or simply $\overline{f_{\mathbf{z}^t}}$ and $\overline{f_{v^t}}$, are the average velocities of scene objects and the observer within $[t, t + \Delta t]$. Note that we use a $z^t$ to capture both the shape and pose information of an object. However, we do not consider shape changes in this work. With the dynamical systems defined, we introduce our assumptions (which defines a tractable subset of all possible situations) as:

- **(A1)** *The high-frame-rate assumption* $\Delta t \to 0$ s.t. $x^{t+\Delta t} \approx x^t$,
- **(A2)** *The large-speed-difference assumption* The data comes from one of two cases (SCFO: Slow Camera, Fast Objects or FCSO: Fast Camera Slow Objects), that satisfy: $|\frac{|\overline{f_{\mathbf{z}}}|}{|\overline{f_v}|}| \geq C_{SCFO}$ or $|\frac{|\overline{f_{\mathbf{z}}}|}{|\overline{f_v}|}| \leq C_{FCSO}$, where $|velocity|$ computes a speed, and $C_{SCFO}$ and $C_{FCSO}$ are positive constants.

**A1** allows us to assume a nearly static scene $\mathbf{z}^t$ or a fixed viewpoint $v^t$ for a short period. Consider an example where we assume a static scene, i.e. $\mathbf{z}^{\tau-\Delta t} \approx \mathbf{z}^\tau \approx \mathbf{z}^{\tau+\Delta t}$, in $[\tau - \Delta t, \tau + \Delta t]$, **A1** essentially allows us to extract $\mathbf{z}^t$ out of a joint sample as: $\mathbf{X}_a = \{(x^t, v^t), \mathbf{z}^t\}_{\tau-\Delta t:\tau+\Delta t}$. An intuitive way to define **A2** is: $|\overline{f_{\mathbf{z}}}| \gg |\overline{f_v}|$ or $|\overline{f_{\mathbf{z}}}| \ll |\overline{f_v}|$, which specify a large speed difference between scene speeds and observer speeds.

These two assumptions enable us to accumulate instant changes (velocities) on one variable (e.g. either $\mathbf{z}^t$ or $v^t$) over a finite number of $\Delta t$ while ignoring the small changes of the other (assumed fixed). We then treat a *slow-camera-fast-objects* (i.e. SCFO) scenario, where $|\overline{f_{\mathbf{z}}}| \gg |\overline{f_v}|$, as an approximate *single-view-dynamic-scene* scenario, and a *fast-camera-slow-objects* (i.e. FCSO) scenario, where $|\overline{f_{\mathbf{z}}}| \ll |\overline{f_v}|$, an approximate *multi-view-static-scene* scenario. Either case allows us to resolve the *temporal entanglement* problem. Importantly, to answer the question: "is a given data sample an SCFO or FCSO sample", we need to quantitatively specify the two assignment criteria $C_{SCFO}$ and $C_{FCSO}$. However, a direct calculation of these two constants is often difficult and does not generalize as: i) $|\overline{f_{\mathbf{z}}}|$ is not available in unsupervised scene representation learning data, and ii) the two constants vary across different datasets. In practice, we cluster the data samples into SCFO and FCSO clusters using only the viewpoint speed $|\overline{f_v}|$, i.e. assuming $|\overline{f_{\mathbf{z}}}| = 1$ for training (see Sec. 3.3). In testing, DyMON treats them equally.

### 3.2 Spatial Object Factorization

DyMON tackles scene spatial decomposition in a similar way to MulMON [31] using a generative model and an inference model. The generative likelihood of a single image observation is modelled with a spatial Gaussian mixture [38, 11]:

$$p_\theta(x^t|\mathbf{z}^t = \{z_k^t\}, v^t) = \prod_{i=1}^{M} \sum_{k=1}^{K} p_\theta(C_i^t = k|z_k^t) \cdot \mathcal{N}(x_{k,i}^t; g_\theta(z_k^t, v^t), \sigma^2\mathbf{I}), \tag{2}$$

where $i$ indexes a pixel location ($M$ in total) and RGB values (e.g. $x_{k,i}^t$) that pertain to an object $k$ are sampled from a Gaussian distribution $\mathcal{N}(x_{k,i}^t; g_\theta(z_k^t, v^t), \sigma^2\mathbf{I})$ whose mean is determined by the decoder network $g_\theta(\cdot)$ (defined in Sec. 2) with trainable parameter $\theta$ and standard deviation $\sigma$ is globally set to a fixed value 0.1 for all pixels. The mixing coefficients $p_\theta(C_i = k|z_k)$ capture the categorical probability of assigning a pixel $i$ to an object $k$ (i.e. $C_i = k$). This imposes a competition over the $K$ objects as every pixel has to be explained by one and only one object in the scene.

DyMON adapts the *cross-view inference module* [31] of MulMON to handle: i) the cross-view object correspondence problem, ii) recursive approximation of a factorized posterior, and iii) temporal evolution of spatial structures (which indicates the major difference between the inference modules of DyMON and MulMON). The decomposition and recursive approximation of the posterior is:

$$p(\mathbf{z}^t = \{z_k^t\}|x^{\leq t}, v^{\leq t}) \approx q_\Phi(\mathbf{z}^t = \{z_k^t\}|x^{\leq t}, v^{\leq t}) = q(\mathbf{z}^0)\prod_t q_\Phi(\mathbf{z}^t|x^t, v^t, \mathbf{z}^{<t}), \tag{3}$$

where $q_\Phi(\mathbf{z}^t|x^t, v^t, \mathbf{z}^{<t})$ denotes the approximate posterior to a subproblem w.r.t. an observation $x^t$ taken from viewpoint $v^t$ at time $t$, and assumes a standard Gaussian $\mathcal{N}(\mathbf{0}, \mathbf{I})$ for the scene prior $q(\mathbf{z}^0)$. The intuition is to treat a posterior inferred from previous observations as the new prior to perform Bayesian inference for a new posterior based on a new observation. We use $\mathbf{z}^t$ to denote the inferred scene representations after observing $x^t$, i.e. a new posterior, and $\mathbf{z}^{<t}$ to denote the new prior before observing $x^t$. Note that we can advance $t$ either regularly or irregularly. The single-view (or within-view) inference is handled by DyMON using *iterative amortized inference* [27] with amortization function $\Phi$ (modelled with neural networks). Refer to Appendix B. for full details about the generative and inference models of DyMON.

## 3.3 Training

To enable DyMON to learn independently the generative relationships between scenes and observations, and viewpoints and observations during training, built upon MulMON's architecture, we break a long moving-cam-dynamic-scene sequence into short sub-sequences (see Algo. 1) where sampling $(x^{t'}, v^{t'}) \sim (x^t, v^t)_{1:T}$ independently of $\mathbf{z}^t$ is possible. Similar to MulMON [31], we then train DyMON by maximizing the following objective function that linearly combines an evidence lower bound (abbr. ELBO) and the log likelihood (abbr. LL) of the querying views:

$$
\begin{aligned}
\mathcal{L} =& \boldsymbol{ELBO} \; + \; \beta \cdot \boldsymbol{LL}_{query} \\
=& \frac{1}{|\mathcal{T}|} \sum_{t \in \mathcal{T}} \mathbf{E}_{q_\Phi(\mathbf{z}^t|\cdot)}[\log p_\theta(x^t|\mathbf{z}^t, v^t)] - \frac{1}{|\mathcal{T}|} \sum_{t \in \mathcal{T}} \mathcal{D}_{\mathrm{KL}}[q_\Phi(\mathbf{z}^t|x^{\leqslant t}, v^{\leqslant t})||q_\Phi(\mathbf{z}^{<t}|x^{<t}, v^{<t})] \\
& + \beta \cdot \frac{1}{|\mathcal{T}| \cdot |\mathcal{Q}|} \sum_{t \in \mathcal{T}} \sum_{t_q \in \mathcal{Q}} \mathbf{E}_{q_\Phi(\mathbf{z}^t|\cdot)}[\log p_\theta(x^q|\mathbf{z}^t, v^q)],
\end{aligned}
\tag{4}
$$

where $\mathcal{T}$ and $\mathcal{Q}$ record the times when DyMON performs inference and $v^t$ interventions (i.e. viewpoint-queried generation) and $\beta$ is the weighting coefficient. We construct $\mathcal{T}$ by sampling $t$ (either regularly or irregularly) with a random walk through $[1, T]$, where a uniform distribution $\mathcal{U}\{\Delta t - 2, \Delta t + 2\}$ of an expected value $\Delta t \, (> 2)$ is used as the step distribution. As shown in Algo. 1, by varying the updating periods of $\mathbf{z}^t$ and $v^t$ (denoted as $\Delta t_{\mathbf{z}}$ and $\Delta t_v$ respectively), DyMON imitates the behaviours of a *multi-view-static-scene* model and a *single-view-dynamic-scene* model to handle the SCFO and FCSO samples respectively. In addition, using different $\beta$ for the SCFO and FCSO samples allows alternating the training focus between spatial reasoning (w.r.t. objects and viewpoints) and temporal updating.

**Assignment Function and Batching** As the samplers of T and Q behave differently for SCFO and FCSO data (see Algo. 1), we need to determine if a $\mathbf{X} \sim \mathcal{D}$ is an SCFO sample or an FCSO sample. Under **A2**, we consider any dataset consisting of only a mix of SCFO and FCSO samples (where a sample is a sequence of images). For a given dataset, we cluster all training samples of a dataset into two clusters w.r.t. the SCFO and FCSO scenarios. This then gives us an assignment function, $\mathbf{assign}(\mathbf{X}; \mathcal{D})$ (as shown in Algo. 1). In practice, to avoid breaking parallel training processes with loading SCFO and FCSO samples into the same batch, we assign the training data beforehand instead of assigning every data sample on the fly during training. This allows to batch FCSO or SCFO samples independently at every training step.

## 4 Related Work

**Single-View-Static-Scene** The breakthrough of unsupervised object discovery based on a primary scenario, i.e. a single-view-image setting, lays a solid foundation for the recent rise of unsupervised object-centric representation learning research. Built upon a VAE [21], early success was shown by AIR [7] that searches for one object at a time on image regions. Because AIR and most of its successors (e.g. [22]) treat objects as flat pixel patches and the image generation process as "paste flat objects on canvas" using a spatial transformer [17], they often cannot summarize well scene spatial properties that are suitable for 3D manipulation: for example, they do not render smaller objects when the objects are "moved" further away from the camera. To overcome this, most recent advances [1, 10, 24, 6, 26, 5] model a single 2D image with a spatial Gaussian mixture model [38, 11] that allows explicit handling of background and occlusions. Although these models suffer from single-view ambiguities like occlusions or optical illusions, they have the potential for attaining

**Algorithm 1: DyMON Training Algorithm**

---

**Input:** training data $\mathcal{D}$

**Hyperparameters** $|\mathcal{Q}|$, $(\beta_{FCSO}, \beta_{SCFO})$, $(\Delta t, \Delta \tau)$ ;    // $\Delta t > \Delta \tau > 2, |\mathcal{Q}| = \mathbf{sizeof}(\mathcal{Q})$

**Initialize** *trained parameters* $\Phi$, $\theta$, *and latent prior* $\boldsymbol{\lambda}^0 = \{(\mu_k = \mathbf{0}, \sigma_k = \mathbf{I})\}$;

**repeat**

    **Sample** *a sequence* $\mathbf{X} = \{(x^t, v^t)\}_{1:T} \sim \mathcal{D}$ ;        // $T$ (RGB images, viewpoints)

    **if** $\mathbf{assign}(\mathbf{X}; \mathcal{D}) == FCSO$ **then**

        $\beta, \Delta t_v, \Delta t_{\mathbf{z}} = \beta_{FCSO}, \Delta \tau, \Delta t$ ;        // $\Delta t_v < \Delta t_{\mathbf{z}}$, update $v^t$ more often

    **else**

        $\beta, \Delta t_v, \Delta t_{\mathbf{z}} = \beta_{SCFO}, \Delta t, \Delta \tau$ ;        // $\Delta t_{\mathbf{z}} < \Delta t_v$, update $\mathbf{z}^t$ more often

    $\mathcal{T} = \mathbf{random\_walk\_t}(\mathrm{s} = 1, \mathrm{e} = T, \mathrm{step\_dist} = \mathcal{U}\{\Delta t_{\mathbf{z}} - 2, \Delta t_{\mathbf{z}} + 2\})$ ;

    $(x, v)$, $t$, $\boldsymbol{\lambda}^t$, $\boldsymbol{ELBO}$, $\boldsymbol{LL}_{query}$, $= \mathbf{X}[1]$, $1$, $\boldsymbol{\lambda}^0$, $0$, $0$;

    **while** $t \leqslant T$ **do**

        $(x^t, v^t) = \mathbf{X}[t]$ ;

        **if** $\mathbf{mod}(t, \Delta t_v) == 0$ **then**

            $v = v^t$ ;                                         // update $v$

        **if** $t \in \mathcal{T}$ **then**

            $x = x^t$ ;                                         // update $x$

            $\boldsymbol{ELBO}^{(t)}, \boldsymbol{\lambda}^t = \mathbf{iterative\_inference}_{\Phi, \theta}(x, v, \boldsymbol{\lambda}^t)$ ;

            $\mathbf{z}^{\mathbf{t}} \sim \mathcal{N}(\mathbf{z}^t; \boldsymbol{\lambda}^t)$ ;                              // sample updated $\mathbf{z}^t$

            $\mathcal{Q} = \{t_q\} = \mathbf{sample\_query\_t}(\mathrm{dist} = \mathcal{U}\{t - \Delta t_{\mathbf{z}}/2, t + \Delta t_{\mathbf{z}}/2\}, \mathrm{size} = |\mathcal{Q}|)$;

            **for** $t_q \in \mathcal{Q}$ **do**

                $(x^q, v^q) = \mathbf{X}[t_q]$;

                $\boldsymbol{LL}_{query} += (1/(|\mathcal{Q}| \cdot |\mathcal{T}|)) \cdot \log p_\theta(x^q | \mathbf{z}^t, v^q)$ ;        // fix $\mathbf{z}^t$, do $v = v^q$

            $\boldsymbol{ELBO} += (1/|\mathcal{T}|) \cdot \boldsymbol{ELBO}^{(t)}$;

        $t += 1$;

    $\mathcal{L} = \boldsymbol{ELBO} + \beta \cdot \boldsymbol{LL}_{query}$ ;                        // $\beta_{FCSO} > \beta_{SCFO}$

    $\theta, \Phi \leftarrow \mathbf{optimizer}_{\mathbf{max}}(\mathcal{L}, \theta, \Phi)$;

**until** $\theta, \Phi$ *converge*;

---

factorized representations of 3D scenes. Our work has close relationship to IODINE [10]: we handle the object-wise inference from an image observation at each time point using the *iterative amortized inference* [27] design and capture the compositional generative process with a spatial Gaussian mixture model.

**Multi-View-Static-Scene** A natural way of resolving single-view ambiguities is to aggregate information from multi-view observations. Although multi-view scene explorations do not directly facilitate object-level 3D scene factorization, Eslami et al. [8] demonstrated that they do reduce the spatial uncertainty and enable explicit 3D knowledge evaluation—novel-view prediction. As combining GQN [8] and IODINE [10], Nanbo et al. [31] showed that MulMON effectively leverages multi-view exploration to extract accurate object representations of 3D scenes. However, like GQN, MulMON can only train on static-scene samples and thus does not generalize well to dynamic scenes ROOTS [3] combines GQN and AIR's merits to perform multi-view-static-scene object-centric representation learning whereas it requires camera intrinsic parameters to overcome AIR's deficiency of 3D scene learning — it is thus camera-dependent hence less general. In our work, we propose DyMON as an extension of MulMON to dynamic scenes and a unified model for unsupervised multi-view object-centric representation learning.

**Single-View-Dynamic-Scene** A line of unsupervised scene object-centric representation learning research was established on the *single-view-dynamic-scene* setting [14, 22, 19], where they explicitly model and represent object dynamics based on video observations. However, as most of these works employ a similar image composition design to AIR, they deal with only flat 2D objects that are similar to MNIST digits and thus cannot model 3D spatial properties. A closely-related work is that of Lin et al. [23], i.e. GSWM, where they modelled relative depth information and pair-wise interactions of

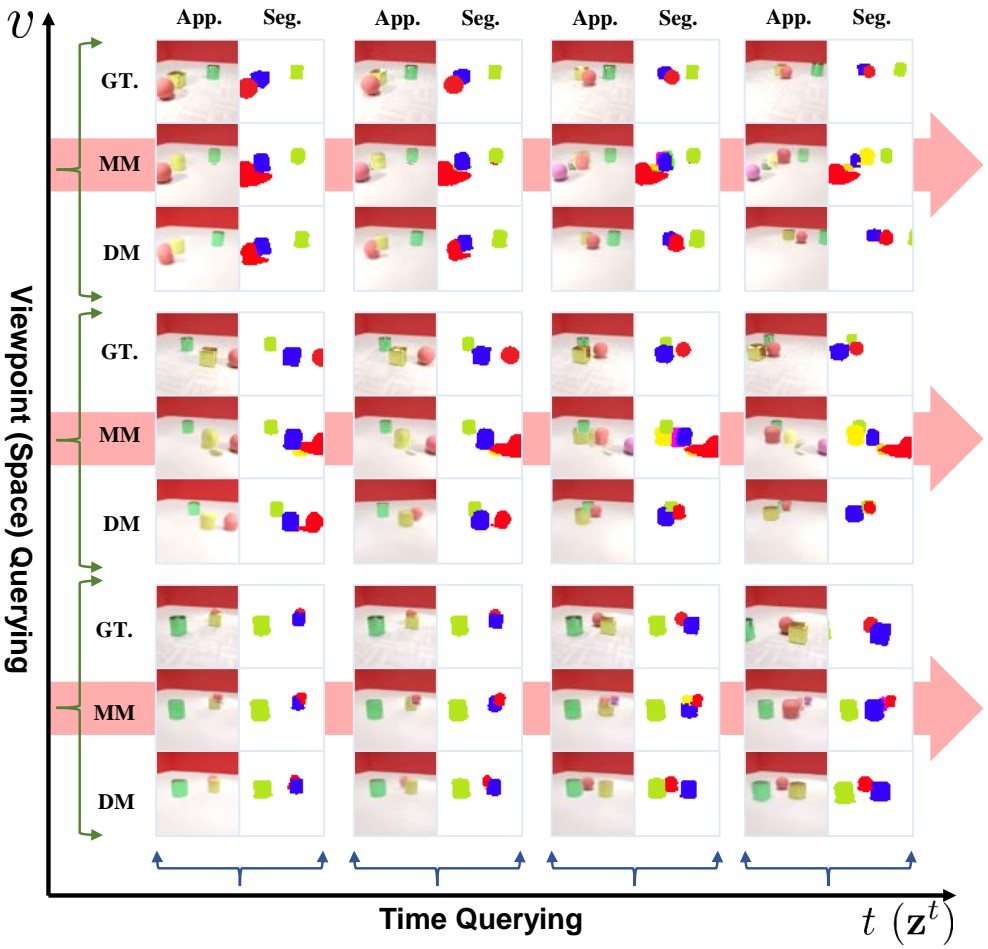

Figure 2: Qualitative results of spatial-temporal factorization. The GT rows show the true scene. The "MM" and "DM" entries are the scene re-rendered from the corresponding models, i.e. MulMON and DyMON respectively. The vertical row pairs show the results from viewpoint changes and the horizontal direction shows the results at different times. Note that we train MulMON and DyMON on different datasets as MulMON cannot train on multi-view-dynamic-scene datasets. We also visualize MulMON's tendency of generating degenerated results along the temporal direction (marked with red arrows).

3D object patches. In our work, the spatial-temporal factorization allows us to show the dynamics and depths of the objects from different viewpoints at different times.

**Other Related Work** As a *multi-view-dynamic-scene* representation learning framework, T-GQN [36] represents the most closely-related work to ours. It models the spatial representation learning at each time step as a stochastic process (SP) and transitions between these time-stamped SPs with a state machine. However, a notable distinction between the problems that T-GQN and DyMON are targeting based on that: 1) T-GQN does not attain object-level scene factorization and 2) a typical T-GQN requires multi-view observations at each time step (as so-called "context") to perform spatial learning so as to get rid of the *temporal entanglement* problem (which has been the core focus of our work). Our work is essentially dealing with disentangled representation learning problems, which are often formulated under the frameworks of causal inference [33, 37, 35] and *independent component analysis* (abbr. ICA) [16, 15]. Unlike traditional disentanglement representation learning works (e.g. [13, 20, 25]) that aims at feature-level disentanglement, in this work, we handle not only the object-level disentanglement that resides in the object-centric representation learning research, but also the time-dependent scene-observer disentanglement problem.Recent trend of neural radiance field (e.g. [29, 28, 34]) are relevant to our work in the sense of 3D scene representations

using multi-view images. However, from an *vision-as-inverse-graphics* [40] perspective, we do not consider them scene understanding models as they only aim to memorize the volumetric structure of a single scene during "training" thus cannot perform representation inference for unseen scenes.

# 5 Experiments

We used two simulated *multi-view-dynamic-scene* synthetic datasets, namely DRoom and MJC-Arm, and a real-world dataset, namely CubeLand (see Appendix C.3 for details), in this work. We conducted quantitative analysis on DRoom and show qualitative results on the other two datasets. The DRoom dataset consists of five subsets (including both training and testing sets): one subset (denoted as DR0-$|\overline{f_z}|$) with zero object motion (*multi-view-static-scene* data), one subset (denoted as DR0-$|\overline{f_v}|$) with zero camera motion (*single-view-dynamic-scene* data), and three *multi-view-dynamic-scene* subsets of increasing speed difference levels from 1 to 3 (denoted as DR-Lvl.1 $\sim$ 3). Each of the five subsets consists of around 200 training sequences (40 frames of RGB images per sequence) and 20 testing sequences (40 frames from 12 different views, i.e. $57.6k$ images). Although DyMON's focus is on a more general problem, we nevertheless compare it against two recent and specialized unsupervised object-centric representation learning methods, i.e. GSWM [23], and MulMON [31], in two respective settings: *single-view-dynamic-scenes*, and *multi-view-static-scenes*. All models were trained with 3 different random seeds for quantitative comparisons. Refer to our supplementary material for full details on experimental setups, and ablation studies and more qualitative results.

## 5.1 Space-Time Querying

The recovery of *the independent generative mechanism* permits DyMON to make both viewpoint-queried and time-queried predictions, i.e. querying across space and times, of scene appearances and segmentations using the inferred scene representations, which enables the below two demonstrations:

**Novel-view Prediction at Arbitrary Times** Recall that a scene observation $x$ is the generative product of a specific scene (composed by objects) and observer at a specific time $t$ with a well-defined generative mapping, i.e. $x = g(\mathbf{z}, v)$ (see sec. 2). Like previous multi-view object-centric representation learning models (e.g. MulMON [31]), we query from an arbitrary viewpoint $v$ w.r.t. a scene of interest $\mathbf{z}$ by fixing $\mathbf{z}$ and manually setting the viewpoint $v$ to arbitrary configurations. Similarly, we can query about the spatial state of a dynamic scene at time $t$ from a specific viewpoint by fixing the viewpoint and manually inputting $z^t$ at arbitrary times $t$ to the generative function. We trained a DyMON on the DR-Lvl.3 data and show qualitatively the prediction results that are queried by space-time tuples in Figure 2.

**Dynamics Replay of Scenes & Objects From Arbitrary Viewpoints** In this experiment, we give DyMON a sequence of image observations of a dynamic scene as input, and have it replay the dynamics from a novel viewpoint using the scene representations it infers from the observations. This is done by fixing the $v$ to the desired values and querying about consecutive times. As the inferred scene representations are factorized in terms of objects, we show in Figure 3 (left) that, besides the complete scene dynamics, DyMON also allows to replay the dynamics of a single object independently of the others. We present the qualitative results on the MJC-Arm datasets in Figure 3 (right) where one can see that DyMON not only replays object dynamics as global position changes, it also captures object local motions.

**Dynamics On Real-World Data** To demonstrate that our model has the potential for real-world applications, we conduct experiments and show qualitative results on real images (i.e. CubeLand data). We refer the readers to Appendix D.4 for the results.

## 5.2 Versatile Evaluation

DyMON is designed to handle object-centric representation learning in a general setting—*multi-view-dynamic-scenes*. In this section, we experiment to evaluate how well DyMON handles the specialized settings.

**DyMON vs. Dynamic Scenes** We first evaluate DyMON's performance in the *multi-view-dynamic-scene* setting in comparison to MulMON. MulMON also learns the *independent generative mechanism* of scene objects and observer, but with a strict static-scene constraint. Note that both DyMON and

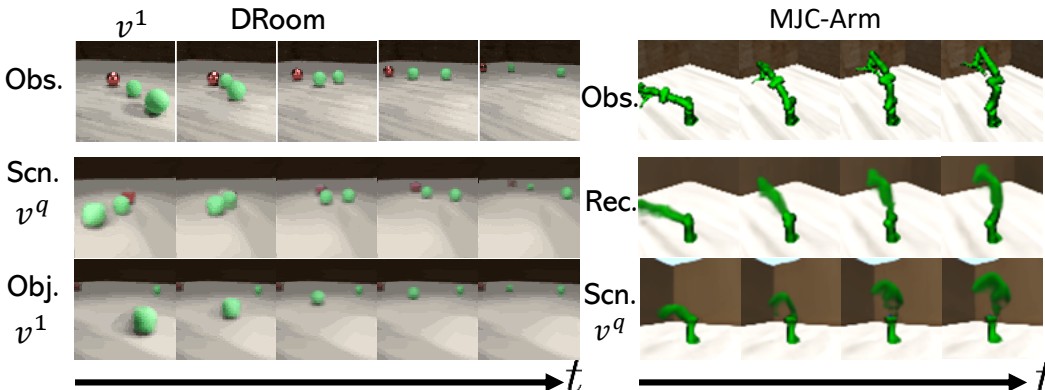

Figure 3: **Left:** DyMON performing dynamics replays on the DRoom dataset, where the first row is the observation sequence input to DyMON, second and third rows show replays of the scene dynamics (all objects' original motions) and object dynamics (just the foreground green ball moves) respectively from an arbitrary viewpoint $v^q$. **Right:** DyMON replays local motions of robot arm from an arbitrary viewpoint (top: observation, middle: reconstruction, bottom: replay from a higher viewpoint).

| Models | MSE↓ | | mIoU↑ | |
| --- | --- | --- | --- | --- |
| | Obs.Rec. | Nv.Obs. | Obs.Seg. | Nv.Seg. |
| MulMON | $0.011 \pm 0.001$ | $\mathbf{0.019 \pm 0.002}$ | $0.511 \pm 0.001$ | $0.461 \pm 0.062$ |
| **DyMON** | $\mathbf{0.004 \pm 0.001}$ | $0.021 \pm 0.002$ | $\mathbf{0.717 \pm 0.000}$ | $\mathbf{0.508 \pm 0.065}$ |

(a) DyMON vs. *Multi-View-Dynamic-Scenes*

| Models | MSE↓ | | mIoU↑ | |
| --- | --- | --- | --- | --- |
| | Obs.Rec. | Nv.Obs. | Obs.Seg. | Nv.Seg. |
| MulMON | $\mathbf{0.006 \pm 0.001}$ | $\mathbf{0.012 \pm 0.005}$ | $\mathbf{0.583 \pm 0.080}$ | $\mathbf{0.538 \pm 0.105}$ |
| **DyMON** | $0.014 \pm 0.001$ | $0.019 \pm 0.007$ | $0.529 \pm 0.005$ | $0.506 \pm 0.105$ |

(b) DyMON vs. *Multi-View-Static-Scenes*

| Models | MSE↓ | mIoU↑ |
| --- | --- | --- |
| | Obs.Rec. | Obs.Seg. |
| GSWM | $0.039 \pm 0.007$ | $0.402 \pm 0.082$ |
| **DyMON** | $\mathbf{0.014 \pm 0.011}$ | $\mathbf{0.682 \pm 0.107}$ |

(c) DyMON vs. *Single-View-Dynamic-Scenes*

Table 1: Quantitative comparisons of DyMON and two baseline models, i.e. GSWM and MulMON, in handling scenarios that the baseline models are specialized at. The models in table (a) are trained and tested on the DR0-$|\overline{f_v}|$ data, and those in (b) and (c) are trained and tested on the DR0-$|\overline{f_\mathbf{z}}|$ data. "Obs." tags reconstructions and segmentations that are computed for the observations and "Nv." tags those from novel viewpoints. Mean ± stddev for 3 training seeds. ↑ indicates higher is better and ↓ indicates the opposite.

MulMON permit novel-view predictions of scene appearances and segmentations, this allows explicit quantification of the correctness and accuracy of the inferred scene representations. We use a mean-squared-error (MSE) measure and a mean-intersection-over-union score (mIoU) measure. We train DyMON on the DR-Lvl.3 subset and MulMON on the DR0-$|\overline{f_\mathbf{z}}|$ subset (because it is UNABLE to train on dynamic-scene data) and conduct comparison across the three DRoom dynamic-scene subsets (i.e. DR-Lvl.1 $\sim$ 3). Table 1a shows that, although we train MulMON on a more strict dataset, i.e. the DR0-$|\overline{f_\mathbf{z}}|$ dataset, DyMON still outperforms MulMON on almost all the various indicators. We show the qualitative comparison results in Figure 2 and observe that MulMON's performance declines along the temporal axis when large object motions appear. As neither DyMON nor MulMON impose any orders for object discovery, we used the Hungarian matching algorithm to find the best match that maximizes the mIoU score to handle the bipartite matching between the output and the Ground-truth masks.

**DyMON vs. Static Scenes** We evaluate how well it handles *multi-view-static-scene* scenarios in comparison with a specialized model, i.e. MulMON. We train and test both DyMON and MulMON on the DR0-$|\overline{f_\mathbf{z}}|$ subset w.r.t. reconstructions and segmentations of both the observed and unobserved views. Table 1b summarizes the results. They show that DyMON can handle this strict constraint

setting, even though it exhibits a slight performance gap compared with the specialized model. Also, it is worth noting that DyMON and MulMON produce high variances in segmentations. One possible reason is that both MulMON and DyMON employ stochastic parallel inference mechanisms that can sometimes infer duplicate latent representations and harm segmentations [32]. This experiment along with the **DyMON-versus-dynamics-scenes** experiment provides useful guidance for model selection in multi-view applications—use a specialized model in a well-controlled environment and DyMON to handle complex scenarios.

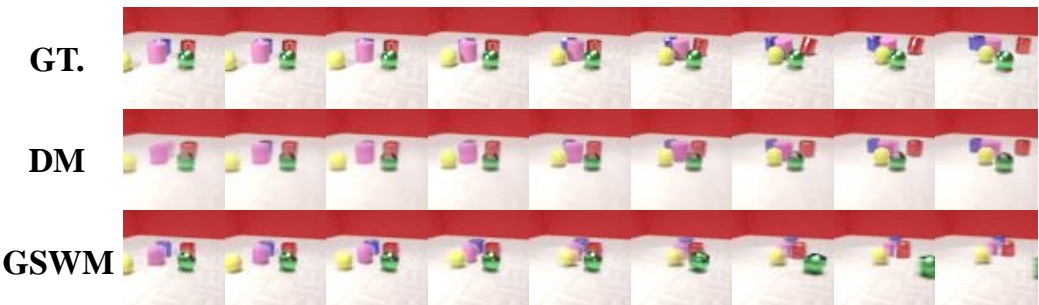

Figure 4: **Left:** Qualitative comparisons of DyMON and GSWM on reconstructing the DR0-$|\overline{f_v}|$ scenes. The GT rows show the actual observations of a dynamic scene, and the "DM" and "GSWM" rows show observation reconstruction results of DyMON and GSWM, respectively.

**DyMON vs. Fixed-View Observations of Dynamic Scenes** We assessed DyMON's performance on handling *single-view-dynamic-scene* observations by comparing it with GSWM [23], which is a specialized object-centric representation model for this specific setting, although it is unable to achieve pixel-level segmentation. We train both DyMON and GSWM on the DR0-$|\overline{f_v}|$ subset and measure the reconstruction quality of the observations. Table 1c shows that DyMON not only outperforms GSWM in observation reconstruction, but it also permits pixel-wise segmentation which the specialized model cannot. The qualitative results in Figure 4 show that GSWM learns better object appearances (especially for textures) than DyMON, whereas DyMON learns more accurate scene dynamics than GSWM. This is understandable as GSWM models object dynamics explicitly, which introduces risks of overfitting the observed motions. DyMON supports well temporal interpolations, i.e. dynamics replays (as shown in Figure 3 and 4), but it does not model the object dynamics nor interactions explicitly like GSWM. As a result, it does not provide readily extrapolatable features along the time (or dynamics) axis for predicting into the future.

**DyMON vs. T-GQN** T-GQN [36] is a closely related work as it targets unsupervised scene representation learning in the multi-view-dynamic-scene settings, even though it does not attain object-centric factorization in the latent space. Although T-GQN requires multi-view observations at each time step (as "context" information) to sidestep the temporal entanglement issue, we nevertheless train it on our DRoom data and show that it fails to represent the DRoom scenes (see Appendix D.3 for the results and discussions).

## 6    Conclusion

We have presented Dynamics-aware Multi-Object Network (DyMON), a method for learning object-centric representations in a *multi-view-dynamic-scene* setting. We have made two weak assumptions that allows DyMON to recover the *independent generative mechanism* of observers and scene objects from both training and testing *multi-view-dynamic-scene* data—achieving *spatial-temporal factorization*. This permits querying the predictions of scene appearances and segmentations across both space and time. As this work focuses on representing the spatial scene configurations at every specific time point, i.e. DyMON does not model dynamics explicitly so it cannot predict the future evolution of scenes, which leaves space for future exploration.

## Acknowledgments and Disclosure of Funding

The first authors would like to acknowledge the School of Informatics, the University of Edinburgh for providing his PhD scholarship. This research is partly supported by the *Trimbot2020* project, which is funded by the European Union Horizon 2020 programme. The authors would like to thank Prof. C. K. I. Williams and Cian Eastwood for valuable discussions.

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
