# OpenReview forum: "Object-Centric Representation Learning with Generative Spatial-Temporal Factorization"
_NeurIPS.cc/2021/Conference — NeurIPS 2021 Poster_

### Official Review · Reviewer_BLv2 · 2021-07-14

**Rating:** 5
**Confidence:** 3

**Summary:**

This paper proposes a method that broadens the scope of multi-view object-centric representation learning to dynamic scenes. The model achieves spatial-temporal factorization by having two weak assumptions: high-frame-rate and large-speed-difference.


**Ethical Concerns:**

No.

**Limitations And Societal Impact:**

Yes.

**Main Review:**

**Strengths:**
1) Overall, this paper is well written, and the technical details are easy to follow.
2) The main idea of unsupervised multi-view object-centric representation learning in the context of dynamic-scene settings is interesting.
3) The factorization of space and time should be important.

**Weaknesses:**

**Assumptions.** The two assumptions of the high-frame-rate and the large-speed-difference are not holding in real-life scenarios. In real synthetic datasets (e.g., GTA and others), the observer is likely moving while the objects in the scene are also performing a movement. I agree the assumptions are valid on the DRoom and MJC-Arm datasets. However, my main concern is that this work would not generalize into real 3D datasets, unlike NRF, which can generalize novel-view synthesis.

**Experiments.**  Table 1a shows that DYMON does not generalize well compared to MulMON[28] on the novel viewpoints. I find it quite surprising. Can the authors explain why that is? Also, why is there a gap between object reconstructions and segmentations?. Additionally, Table 1b is not clear to me. Why does the MulMON model perform better than DyMON? I would expect from a model that learns dynamics to generalize over static scenes.


My main concern is I cannot see how this work generalizes over real 3D datasets and helps the field to move forward. I would expect the authors to add at least one real 3D dataset (or at least more realistic environments) since the dynamics play a critical factor in real-life scenarios more than on toy tasks. Furthermore, generalization and understanding of objects and observers' movements are vital for multiple tasks and domains, and thus I believe the spatial-temporal factorization should be leveraged on more realistic scenarios. On the other hand, I liked this paper, its contribution, and its motivation. I am open to the authors' feedback and other reviewers' opinions.



**Time Spent Reviewing:**

6-8

---

> ### Author Response · Authors · 2021-08-10
> **Response to Reviewer BLv2 (We have added real-data results)**
>
> We thank the reviewer for the valuable review and feedback. Before answering the reviewer's questions we would like to clarify that 1) DyMON can train on both static-scene data and dynamic-scene data while MulMON can only train on static-scene data where it can easily learn spatial reasoning ability as the *temporal entanglement* problems do not exist, and 2) the DyMON and MulMON models we used to produce the Table 1a results are trained on two different datasets as MulMON cannot train on the DyMON dataset. The experiments in Sec. 5.2 are only to analyse how well DyMON, a model that handles a more general set of problems, handles all other specialised settings.
>
>
>
> > *"Table 1a shows that DYMON does not generalize well compared to MulMON[28] on the novel viewpoints. Can the authors explain why that is?"*
>
> Here we first correct the standard deviations for the Table 1a results that is referred to:
>
> | Models | Nv.Obs. (MSE)              |
> | ------ | -------------------------- |
> | MulMON | $\mathbf{0.019 \pm 0.002}$ |
> | DyMON  | $0.021 \pm 0.002$          |
>
> Although the results suggests a better performance of MulMON than DyMON on novel-view synthesis, a $1-stddev$ gap is not statistically significant enough to be a strong support to the hypothesis. The same situation applies to the novel-view segmentation results too---DyMON and MulMON produce close novel-view segmentation prediction results (i.e. low statistical significance). So it remains unclear whether if it is the randomness introduced in the training or evaluation that causes this. Despite the low statistical significance, we admit that there is indeed a possibility that MulMON has slightly better spatial reasoning ability from the multi-view-static-scene data on which it was trained. As the multi-view-static-scene datasets are well-constrained and free of camera motions' interference (i.e. temporal entanglement), it is easier to train a model on these datasets than DyMON datasets.
>
>
>
> > *" Table 1a shows...why is there a gap between object reconstructions and segmentations?"*
>
> If we understand the reviewer's question correctly, the reviewer is asking why MulMON and DyMON produce different comparison results on novel-view observation predictions and segmentation predictions. This is because the input variables to the MSE  and mIoU computations are different: the MSE computation takes in the generator output $g_{\theta}(\{z_k\}) \in \mathbb{R}^{H\times W \times K \times 4}$, i.e. 4-channel output $RGB\alpha$, while mIoU takes in only the 1-channel $\alpha \in \mathbb{R}^{H\times W \times K \times 1}$. Therefore, the performance gap between object reconstructions and segmentations is understandable as the MSE and mIoU are dependent of different variables.
>
>
>
> > *"Table 1b is not clear... Why does the MulMON model perform better than DyMON? I would expect from a model that learns dynamics to generalize over static scenes."*
>
> We clarify that DyMON does not learn dynamics at all. This is because both DyMON and MulMON used to produce Table 1b were trained on the DR0-$|f_{\mathbf{z}}|$ dataset (i.e. a moving-camera-static-scene dataset). In this case, DyMON did not learn the ability that is necessary to handle scene dynamics as there is no object dynamics in the training data.
>
> For the performance gap between DyMON and MulMON in Table 1b, we consider it is the training procedure (see Algorithm 1) of DyMON that hinders its learning of spatial reasoning ability. DyMON's training procedure breaks down a long sequence (recorded along a long camera trajectory) into multiple sub-sequences and performs the $do(v^t=v^q)$ operations to acquire view reasoning ability on the sub-trajectories. The drawback of this is the view changes that the $do(v^t=v^q)$ operations can cover is limited by the hyperparameter $\Delta t_{\mathbf{z}}$, so it is understandable DyMON does not reason well when dealing with wide-range viewpoints. For this, we refer the reviewer to Figure 2 of the main paper, where the difference between the ground-truth and the predicted observations increases as we increase the view changes (go up along the vertical axis). On the other hand, MulMON always performs the  $do(v^t=v^q)$ operations on the entire sequence during training so it learns to better handle wide-range novel viewpoints and produce better results.
>
>
>
> > *"I would expect the authors to add at least one real 3D dataset "*
>
> It is concluded in [1] that the power of generative models is limited by the current hardware. For the same reason, we chose simple objects in this work to avoid heavy computations. In theory, this line of research can generalise well on more realistic scenes as they do not make any assumptions about the scenes and objects. The Sketchy real-world datasets used in [2] is probably the most "realistic" data that we have seen in this line of research.
>
> Nonetheless, to show that DyMON does generalise to real-world data, we have conducted an experiment of DyMON performing on a real-world multi-view-dynamic-scene dataset that we recorded. We have created a video demo to demonstrate DyMON's performance. We will update the final paper to include these results. For now,  the reviewer can check the video demo (```demo_real_world_data.mp4```) via the following link: https://figshare.com/s/1e21452856b7744f2dde.
>
> [1] Eslami et al. Neural scene representation and rendering. Science, 2018
> [2] Engelcke et al. GENESIS-V2: inferring unordered object representations without iterative refinement. arXiv, 2021

---

> > ### Author Response · Authors · 2021-09-06
> > **Any follow-up questions/concerns?**
> >
> > We thank the reviewer again for the valuable suggestions. As it is getting to the end of the discussion period, we would like to ask if the reviewer has any follow-up questions/concerns that we can further clarify before the window closes?

---

### Official Review · Reviewer_dvA3 · 2021-07-15

**Rating:** 6
**Confidence:** 3

**Summary:**

This paper proposes DyMON, an unsupervised object-centric representation learning model that can be applied to scenes with dynamic objects and multiple viewpoints. DyMON is trained to obtain object latents that are independent of the viewpoints. The model builds off of MulMON, extending it to work with scenes with dynamic objects. Two assumptions are made about the training scenes: 1) a high frame rate assumption and 2) the relative speed difference between the camera and the objects is large -- that is, either the camera moves much faster than the objects or the objects move much faster than the camera. With these assumptions, the problem can be reduced to either a multi-viewpoint, static objects problem or a single-viewpoint, dynamic objects problem. Through experiments, the authors demonstrate that DyMON can be used to query predictions of the scene at arbitrary times and viewpoints.


**Limitations And Societal Impact:**

Yes

**Main Review:**

This paper is well-motivated, investigating the problem of learning object-centric representations in scenes with dynamic objects and multiple viewpoints without any supervision. This is an area previous works (that I know of) have not explicitly explored and is very important for modeling more realistic scenes.

The authors make several assumptions about the dataset and given those assumptions, the approach they take is reasonable and sound. However, while the authors claim to "resolve" (L127) the temporal entanglement problem, it seems that by making these assumptions, they have instead sidestepped the problem by restricting the dataset to a domain where the problem is not significant.

The experiments are encouraging, especially the ones querying novel viewpoints and times. However, I do not believe the DyMON vs. Dynamic Scenes section is very informative since MulMON is trained on a different dataset than it is evaluated on. I would also be curious how GSWM performs on the multi-view-dynamic-scene dataset. Although it does not explicitly take into account camera viewpoint, it can handle dynamic backgrounds (ie. context), so it may be able to model these scenes. Lastly, in addition to reporting MSE, it would be interesting to see comparisons of the reconstructions of DyMON vs. the baselines to see where the different models qualitatively fail.

My main concern with the paper is with the novelty. If I understand correctly, the model is almost identical to MulMON, but the difference is in the training procedure. While the DyMON training algorithm allows the model to factorize the camera dynamics from the object dynamics, this combined with the aforementioned assumptions made on the dataset limits the contributions of this paper. I encourage the authors to clarify the novelty of their approach over MulMON. I am willing to change my score if I have missed something.

Minor issues / typos:

- GSWM seems to be cited incorrectly (SPACE is cited twice)
- In Equation (4), I'd recommend something other than q for the query since it is overloaded.
- L23: These method -> These methods
- L153: veiw -> view
- L165 -> "we have" seems not necessary


**Time Spent Reviewing:**

12

---

> ### Author Response · Authors · 2021-08-10
> **Response to Reviewer dvA3**
>
> We thank the reviewer for the valuable review and feedback. Here we address the reviewer's main concern first, i.e. the novelty of DyMON over MulMON, and then respond to the reviewer's other comments.
>
> > *"the model is almost identical to MulMON, but the difference is in the training procedure...clarify the novelty of their approach over MulMON"*
>
> First of all, we believe that, in general, the need for a novel approach is the consequence of a novel problem rather than the cause. In our case, as we attempt to tackle a novel problem in the context of object-centric representation learning, we propose DyMON to handle the *temporal entanglement* problem which represents the major technical challenge of going beyond the static-scene assumption that is made in other multi-view object-centric learning prior works (e.g. MulMON). Specifically, as discussed in Sec 2 of the main paper, we need to sample $(x^{t'}, v^{t'}) \sim \{(x^t, v^t )\}_{1:T}$ independently of $\mathbf{z}^t$ from $\{(x^t, v^t , \mathbf{z}^t)\}_{1:T}$. So we built upon MulMON's architecture and tackle directly this *temporal entanglement* challenge by proposing: 1) a novel training procedure that breaks a long moving-cam-dynamic-scene sequence into short sub-sequences (see Algorithm 1) where the aforementioned sampling is possible (with the SCFO and FCSO approximations), and this leads to 2) a novel approximation of the evidence lower bound (see Equation 4)---the formations and formulations of $\mathcal{T}$ and $\mathcal{Q}$  (see Algorithm 1) defines a novel training objective function.
>
>
>
> > *"while the authors claim to "resolve" (L127) the temporal entanglement problem, it seems that by making these assumptions, they have instead sidestepped the problem by restricting the dataset to a domain where the problem is not significant...assumptions made on the dataset limits the contributions of this paper"*
>
> We agree with the reviewer that DyMON has limitations---in fact, this holds true for all the models. However, we don't consider DyMON sidesteps the *temporal entanglement* problem by restricting the datasets to mild cases. We list three reasons as follow:
>
> 1) The *temporal entanglement* defines the major technical challenge of going beyond the static-scene assumption that is made in other multi-view object-centric learning prior works (e.g. MulMON). As the first attempt that tackles this novel problem in the context of object-centric representation learning, we proposed DyMON to close the gap between the *static-scene* and the *dynamic-scene* setups. Therefore, DyMON is the first (but not the final) attempt at resolving the *temporal entanglement* problem from this point of view.
> 2) The content that the reviewer referred to (i.e. line 127 or Sec. 3.1) is to highlight the high-level idea behind DyMON. We refer to Algorithm 1 of the main paper for the actual training design of DyMON. DyMON approximates a long moving-cam-dynamic-scene sequence with multiple short segments or sub-sequences on which the high-level idea (i.e. SCFO and FCSO approximations) can be applied. As most video datasets satisfy our assumption A1, we believe that A2 is actually a weak assumption as one can always subdivid difficult sequences into simpler ones (by setting different $\Delta \tau$ and $\Delta t$ in Algorithm 1) to further ease the violation of A2. I.e. the approximation is not to sidestep the problem, but to solve the *temporal entanglement* problem by dividing and conquering its sub-problems. So whether it sidesteps each of the micro sub-problems nor not, it does resolve the macro *temporal entanglement* problem.
> 3) To better understand how robust is DyMON against the violation of our assumption A2, we did conduct  experiments to analyse how DyMON performs against 3 levels of speed differences between the camera and objects. The results of this has been included in the appendix Sec. 4.1 as *"Assumption Validation"*, where we trained and tested DyMON on 3 different datasets (of 3 different levels of A2 violations), i.e. the DR-Lvl.1 ∼ 3 datasets (see line 230-231 in the paper and Sec. C.1 in the appendix). We see that increasing violation of our assumption A2 does indeed cause performance drop, specifically on DR-Lvl.1 where the problem is significant (see Figure 2 in the appendix), but the drop is not drastic. This shows that our model is robust against different levels of datasets restrictions. We will explore more levels ($\ge 3$) of camera-object speed differences to clarify the message and further identify the limitations.
>
>
>
> >  *" how GSWM performs on the multi-view-dynamic-scene dataset...Although it does not explicitly take into account camera viewpoint, it can handle dynamic backgrounds (ie. context), so it may be able to model these scenes."*
>
> We thank the reviewer for the great suggestion. We will update *Table 1(a)* include the results in final paper once the training is finished. It is also worth noting that the comparison results will NOT affect any of the major claims that we made.
>
>
>
> > *"in addition to reporting MSE, it would be interesting to see comparisons of the reconstructions of DyMON vs. the baselines to see where the different models qualitatively fail."*
>
> We have created several qualitative comparison samples and will have them included in the final paper. For now, the reviewer can check them via the following link: https://figshare.com/s/1e21452856b7744f2dde.

---

> > ### Comment · Reviewer_dvA3 · 2021-08-30
> > **Updated Score**
> >
> > Thank you for taking the time to run additional experiments and address several of my concerns. I have decided to increase the score given the clarification you have provided. I would recommend including some of the discussion here in the final paper as it will improve the clarity of the paper.

---

### Official Review · Reviewer_KAQi · 2021-07-15

**Rating:** 6
**Confidence:** 5

**Summary:**

This work proposed a Dynamics-aware Multi-Object Network (DyMON), a model that captures object-centric representations in dynamic settings and enables querying novel viewpoints as well as single objects from the scene. DyMON builds on top of MulMON, a method that works in multi-view static settings.

**Main Review:**

Strengths:
1) The problem setting is novel and has not been tackled from an object-centric representation point of view.

Weaknesses:

2) Is assumption A2 i.e having a large-speed difference between the camera and object realistic? Or is this just a trick to make the model learn? If so, then there must be some results shown on a dataset wherein both scene velocity and observer's velocity are perceivable.

3) Baselines: The baselines used to compare are not fair. For example, T-GQN [Ref1] is a closely related work. This captures the model's dynamics in multi-view dynamic scenes (though not object-centric). It is surprising that the author(s) have missed this.

4) Related works: Another closely related work for multi-view static setting is that of ROOTS [Ref2] which should be discussed in the related works section.

5-a) Experiments: The author(s) claim that DyMON "permits pixel-wise segmentation which the specialized model (GSEWM) cannot". This is false. There are object-level masks in GSWM that essentially are the object segmentation masks. So, I don't see a reason why mIoU is N/A in Table 1(c).

5-b) From my experience, GSWM has a very low MSE (in static view dynamic object scenes). Given that the DRoom dataset is similar to CLEVERER used in GSWM, I expect the reconstructions to be good. Can the author(s) provide the reconstruction comparison (either in the main paper or in the appendix)?

6) Qualitative Results: Comparison against MulMON or GSWM in the qualitative results needs to be shown to visualize where DyMON is doing better. I'd urge the author(s) to include the qualitative comparison and give insights into where other models fail.

Suggestions:
7) The testing algorithm can be included in the main paper to make things clear. $\beta_{\text{FCSO}}$, $\beta_{\text{SCFO}}$ makes the reader think that they are also inferred during the test time, which is not the case.

8) As mentioned in point 6, figures 2 and 3 need to improve. Fig. 2 occupies too much space (can reduce the number of examples and instead put MulMon and GSWM results).

9) Abbreviations for Nv. Seg and Nv. Obs needs to be expanded in the caption of Table 1.

Other minor details (not considered for decision):
10) The paper seems to have some typos, please fix them. Otherwise in general the paper is well written.

References:

[Ref1] Sequential neural processes, G Singh, J Yoon, Y Son, S Ahn - In NeurIPS, 2019

[Ref2] ROOTS: Object-Centric Representation and Rendering of 3D Scenes, Chang Chen, Fei Deng, Sungjin Ahn, JMLR, 2021


Justification for Rating: Lack of proper baselines , comparisons with other methods and some confusion in the assumptions of the model are the reason why I'm currently giving the paper a  reject. I'd be looking forward for the rebuttal of the author(s) and change my score accordingly.

-------

Edit 1 (post rebuttal): The author(s) have addresses some of my concerns and I thank them for it. Hence, I've decided to increase my rating to "5: Marginally below the acceptance threshold" but cannot increase any further due to lack of proper baseline.

----
Edit 2 (post rebuttal and post T-GQN expt.) : Thanks to the author(s) for including the T-GQN baseline. Having addressed my concerns I'm increasing to the rating to a Weak accept. As mentioned earlier, the problem direction is quite novel and the proposed method is a good step in this direction, I'd hope more works come out which relax the assumptions and/or work in a much more realistic scenarios.

**Time Spent Reviewing:**

8 hours

---

> ### Author Response · Authors · 2021-08-10
> **Response to Reviewer KAQi**
>
> We thank the reviewer for the valuable review and feedback.
>
> > *"Is assumption A2 i.e having a large-speed difference between the camera and object realistic?...there must be some results shown on a dataset wherein both scene velocity and observer's velocity are perceivable"*
>
> We agree with the reviewer that our assumption A2 is not always true in real life, neither is the *static-scene* assumption made in MulMON. As the first attempt that tackles this novel problem in the context of object-centric representation learning, we proposed DyMON to close the gap between the *static-scene* setups and the *dynamic-scene* setups. To better understand how robust is DyMON against the violation of our assumption A2, we did conduct  experiments to analyse how DyMON performs against 3 levels of speed differences between the camera and objects. The results of this has been included in the appendix Sec. 4.1 as *"Assumption Validation"*, where we trained and tested DyMON on 3 different datasets (of 3 different levels of A2 violations), i.e. the DR-Lvl.1 ∼ 3 datasets (see line 230-231 in the paper and Sec. C.1 in the appendix). In order to improve the clarity, we will explore more levels of speed differences (violations of A2) in the final paper.
>
>
>
> >  "*T-GQN [Ref1] is a closely related work* "
>
> We thank the reviewer for recommending this T-GQN work, which seems to be a good baseline of DyMON, even though it's not an object-centric model.  Having the comparison between T-GQN and DyMON on both observation reconstruction and novel-view prediction qualities will make an improvement to the paper. We will include the results in the final paper once the training of T-GQN is finished. It is also worth noting that the comparison results will not affect any of our claims.
>
>
>
> > "*Experiments: 'DyMON permits pixel-wise segmentation which the specialized model (GSEWM) cannot'. This is false. There are object-level masks in GSWM that essentially are the object segmentation masks. So, I don't see a reason why mIoU is N/A in Table 1(c).* "
>
> Although attaining holistic image segmentations is not claimed by the original paper nor included in the official implementation, the reviewer is correct that we can obtain GSWM segmentation masks by post-processing the intermediate object-level attention maps of GSWM. We have fixed the incorrect statement and updated Table 1c in the paper. Table 1c is now:
>
> | Models | Obs.Rec. (MSE)             | Obs.Seg. (mIoU)            |
> | ------ | -------------------------- | -------------------------- |
> | GSWM   | $0.039 \pm 0.007$          | $0.402 \pm 0.082$          |
> | DyMON  | $\mathbf{0.014 \pm 0.011}$ | $\mathbf{0.682 \pm 0.107}$ |
>
>
>
> >  "*Qualitative Results: Comparison against MulMON or GSWM in the qualitative results* "
>
> This is additional content for the paper, but we have created several qualitative comparison samples and will include them in the final paper. For now, the reviewer can check them via the following link: https://figshare.com/s/1e21452856b7744f2dde.

---

> > ### Comment · Reviewer_KAQi · 2021-08-24
> > **some minor questions before final decision**
> >
> > I thank the author(s) for their rebuttal and for sharing anonymous qualitative comparison link as well. I do have a couple of follow up questions regarding the work:
> >
> > 1) In the figure shared, the reconstruction comparison between DyMON vs GSWM and DyMON vs MulMON are for Tables 1(c) and 1(b) respectively. Is it possible to have a look at reconstruction in case of multi-view dynamic scenes (table 1(a)) as well? This would be more interesting part since DyMON is tackling such scenarios. Also please mention your observations on how each model fails on the corresponding scene. For example: GSWM predicts collision incorrectly because of which the green ball seems to go out of the scene even though it is not hit by the yellow ball.
> >
> > 2) If there is any update on preliminary results of T-GQN vs DyMON, I'd request the author(s) to share them. Without the baseline it'd be difficult to judge this work.
> >
> > 3) I'd disagree with the author(s) statement *"Although attaining holistic image segmentations is not claimed by the original paper nor included in the official implementation"*. Eq 53 of G-SWM paper and [this line](https://github.com/zhixuan-lin/G-SWM/blob/1423553c30e4340cbe2278639316941cfd16acb6/src/model/gswm/fg.py#L892) in the code precisely does project the mask from *glimpse* to *image* level. But, I do think that the author(s) have understood my point correctly and I appreciate that the comparison has been shown in the rebuttal.
> >
> > As of now I haven't updated my score and would be waiting for author(s) and then edit my final rating (in case it changes).

---

> ### Author Response · Authors · 2021-09-03
> **Perhaps the lack of other comparison approaches is a consequence of the novelty of exploring factored implicit representations suitable for modelling dynamic scenes.**
>
> We have provided comparisons between DyMON and two baselines, i.e. MulMON and GSWM, in the paper. We have also added the requested qualitative comparison results to the anonymous website and will add them to the supplementary materials. However, we are uncertain what could be an improved baseline from the reviewer's perspective. As suggested we trained T-GQN using the standard settings and it performed badly on the DRoom datasets (as is shown in the anonymous upload site and
> as will be shown in the final supplementary materials). Importantly, NONE of the aforementioned baselines (i.e. MulMON, GSWM, and T-GQN) explores unsupervised representation learning in the *multi-view-dynamic-scene* settings from an *object-centric* point of view, and we are not aware of further approaches to compare beyond those already in the paper and T-GQN.
>
> Perhaps the lack of other comparison approaches is a consequence of the novelty of exploring factored implicit representations suitable for modelling dynamic scenes. Our results show top performance in this area so far.

---

### Official Review · Reviewer_zpN5 · 2021-07-16

**Rating:** 6
**Confidence:** 3

**Summary:**

The following work proposes a method for learning the factorized latent structure of a scene. Specifically, while the primary prior work MulMon [28] assume static objects while the viewpoint varies through time, this work allows for both camera and object motion within the training data. To achieve their goal of factorizing object motion from camera motion, they assume high framerate and use clustering to determine cases where the camera is moving quickly and the objects slowly, versus cases where the camera moves slowly but the objects move quickly. Notably, once trained, they are able to resynthesize the scene's dynamic objects from novel viewpoints.

**Limitations And Societal Impact:**

As stated in my main review, I don't think the technical limitaions are adequately addressed. I don't perceive any potential negative societal impact.

**Main Review:**

Strengths:
- The ability to resynthesize dynamic scenes from novel viewpoints is impressive
- Using clustering to identify cases of fast object slow camera and slow object fast camera seems like a highly practical solution to a difficult problem

Weaknesses/Clarity concerns:
- I think there's a lot of critical detail in the appendix, making the paper a bit difficult to follow. Specifically, I don't think I fully follow the details on how the data clustering for SCFO and FCSO works
- There's a fair amount of prior art being left out here. Specifically there's a lot of works in unsupervised 2D landmark learning that are formulated very similarly.  [1], for example similarly disentangles dynamic objects from videos, falling under the "slow camera fast object" category specified in this work. [2] is another related work in the multi view setting.
- I believe there are a number of additional unstated limitations that should be made clear here. What type of objects can we model here? By using spatial Gaussian mixtures, this typically assumes that the objects can be decomposed into rigid approximately convex shapes.
- It appears that this model is capable of modeling the background as well. How do the spatial gaussians capture the background, considering the background spatially contains the foreground objects, and I assume the gaussians should not overlap too much?
- What happens if the number of mixture components exceeds the number of objects by a lot? What happens if it's too small?

Overall, I find the work pretty interesting despite the limitations. While it's quite possible that I've missed some critical details during my read-through, I would appreciate if the authors could point me to where the information I missed was, or clarify in their rebuttal.


[1] Lorenz et al. Unsupervised Part-Based Disentangling of Object Shape and Appearance. CVPR 2019

[2] Rhodin et al. Neural Scene Decomposition for Multi-Person Motion Capture. CVPR2019

Post rebuttal:
I think the authors have adequately addressed most raised concerns and points of confusions. I agree with some other reviewers that the experimental setting is a bit too simple to be convincing, but I think the overall task is novel and the results are pretty interesting. For this reason, I maintain my original rating.

**Time Spent Reviewing:**

5

---

> ### Author Response · Authors · 2021-08-08
> **Response to Reviewer zpN5**
>
> We thank the reviewer for the valuable review and feedback.
>
> > *"how the data clustering for SCFO and FCSO works"*
>
> The data clustering is only for preprocessing the training data as we consider any multi-view-dynamic-scene dataset consisting of only a mix of SCFO and FCSO samples, i.e. sampled from two sub populations of the training set. This is true if our assumption A2 holds. This allows us to cluster training samples into two clusters based on the average camera speeds of the sequences. (see line 127-133 in the paper).
>
> > *"prior art being left out... unsupervised 2D landmark learning"*
>
> The reviewer is correct that there are prior works, which are relevant in terms of application scenarios, and which are left out and we are aware of that. The object-centric representation learning literature covered in this paper is in the context of generative approaches, where the goal of these generative approaches is to identify the underlying generative mechanism of the observations (see line 63-77 in the paper). In other words, instead of extracting object information from the observation like non-generative methods do, generative methods aim to answer a fundamental question: what scene (object configurations) produce the observations. The benefits of generative approaches are that they, in general, make fewer assumptions than the non-generative approaches, e.g. do not require 1) an input background image (as in the suggested NSD paper), nor 2) an implicit 2D transformation assumption so an intermediate and potentially pre-trained spatial transformer network won't fail (as in the suggested part-based disentanglement paper). Nonetheless, we will include the suggested references in the *"Other Related Work"* part of Sec.4 to clarify the differences.
>
> > *"What type of objects can we model here? By using spatial Gaussian mixtures, this typically assumes that the objects can be decomposed into rigid approximately convex shape"* ... *"How do the spatial gaussians capture the background?...the gaussians should not overlap too much"*
>
> DyMON, as well as prior works in this line of research (such as MulMON [1], MONet [2], and IODINE [3]), do not make any explicit assumptions about the shapes or objects. The background component is treated the same as the foreground objects, which is also free of any prior geometric assumption (e.g. overlaps in the image space). In theory, the objects can be anything such as the ducks and mugs examples in MulMON (upon which this work is built) the use of simple shapes here is only a computation-friendly choice in terms of generative models' training.
>
> To clarify, the attributes of an object, including 3D properties like shapes and poses, are captured in the latent space by one of the $K$ *independent* Gaussians. Specifically, each object in a scene is statistically modelled by a single latent Gaussian. Therefore, besides the statistics assumption that latent object representations are samples of Gaussian family, there is not any geometric assumption made. To render an image, the $K$ object representations will be mapped to the image space by the decoder/generator to produce $K$ *independent* object layers. As shown in Eq.2, the mixing distribution of the $K$ *independent* object layers is defined at each pixel location indicating the assignment probabilities of the pixel to the $K$​ *independent* object layers. In other words, they captures the visibility/occluding relations of the scene objects w.r.t. a observation angle.
>
> [1] L. Nanbo et al. Learning object-centric representations of multi-object scenes from multiple views, NeurIPS 2020
> [2] C. P. Burgess et al. MONet: Unsupervised Scene Decomposition and Representation, 2019
> [3] K. Greff et al. Multi-Object Representation Learning with Iterative Variational Inference, ICML 2019
>
> > *"What happens if the number of mixture components exceeds the number of objects by a lot? What happens if it's too small?"*
>
> This line of works do not assume a known number of scene objects. However, a general assumption made in this line of work is that the number of object slots (components), i.e. $K$​, is greater than or equal to the number of scene objects in a scene (see line 141-143 in the paper). Although breaking the assumption by setting an insufficient $K$​ will likely lead to wrong grouping of objects, the assumption is a weak assumption and can be easily satisfied by setting the number of object slots to a larger integer, e.g. $50$​. In this line of works, if the number of components exceed the actual number of scene objects, then the redundant components (modes) will "collapse" --- capture nothing (see Figure 3 in [1] and  Figure 9 in [2] for examples). This is expected and will not affect the rendering/reconstruction as the mixing probabilities of these components (in the Gaussian mixture) will be small such that their mixing effects w.r.t. image rendering will be suppressed. However, setting a large $K$​ will increase computational cost or even cause our-of-memory issues. Therefore, in practice, we often set $K$​ using a large and "affordable" integer.

---

> > ### Comment · Reviewer_zpN5 · 2021-08-27
> > **Some additional questions**
> >
> > Thank you for clarifying!
> >
> > I clearly misunderstood the original description, but would be be accurate to say that the 'spatial' gaussian mixture lies in the 3 dimensional RGB domain as opposed the the spatial coordinate domain? In that sense the approach is much more related to gaussian-mixture-based image segmentation approaches than those that I listed.
> >
> > As an additional note, I think notation for the gaussian might be off? $x^t$ does not appear anywhere on the right hand side of the formula. I would assume you need something like $\mathcal{N}(x_{i}^t | g_\theta(z_{k}^t, v^t), \sigma^2I)$ to get the likelihood of a pixel's RGB value.

---

> > > ### Author Response · Authors · 2021-08-27
> > > **We will include the suggested changes for further clarification**
> > >
> > > > *"the 'spatial' gaussian mixture lies in the 3-dimensional RGB domain as opposed the spatial coordinate domain...more related to gaussian-mixture-based image segmentation approaches"* *
> > >
> > > Yes, the reviewer is correct and we will add one sentence to clarify this in the final paper.
> > >
> > > >  *"notation for the gaussian might be off... would assume you need something like $\mathcal{N}(x^t | g_{\theta}(z^t_k, v^t), \sigma^2 I)$  to get the likelihood of a pixel's RGB value"*
> > >
> > > We thank the reviewer for the suggestion and will update the notion to $\mathcal{N}(x^t ; g_{\theta}(z^t_k, v^t), \sigma^2 I)$ in the final paper.

---

### Author Response · Authors · 2021-08-10
**Overall response to all reviewers**

We would like to thank all reviewers for their thorough reviews and helpful feedback/suggestions on this work, which, to the best of our knowledge, is the first (but not the final) attempt to tackle unsupervised object-centric representation learning in the multi-view-dynamic-scene settings. The majority of the issues raised by the reviewers can be dealt with easily and included in the final paper.

---

### Decision · Program_Chairs · 2021-09-27

**Decision:**

Accept (Poster)

**Comment:**

- The proposed method is tackling an important novel problem. The reviewers found the approach is reasonable and sound but technically not very novel.
- Many concerns (including T-GQN) from the reviewers are addressed well by the rebuttal. These results should be included in the updated version.
- The strong assumptions the method takes to work are the main limitations of the approach but as the first work in the direction I think it still has some contribution.
- Importantly, some key related works like T-GQN and ROOTS are missing and should be discussed with enough emphasis in the revision.